# 3D Cell Culture Models in COVID-19 Times: A Review of 3D Technologies to Understand and Accelerate Therapeutic Drug Discovery

**DOI:** 10.3390/biomedicines9060602

**Published:** 2021-05-26

**Authors:** Guadalupe Tonantzin de Dios-Figueroa, Janette del Rocío Aguilera-Marquez, Tanya A. Camacho-Villegas, Pavel H. Lugo-Fabres

**Affiliations:** 1Department of Medical and Pharmaceutical Biotechnology, Centro de Investigación y Asistencia en Tecnología y Diseño del Estado de Jalisco A.C. (CIATEJ), Av. Normalistas 800, Colinas de las Normal, Guadalajara, Jalisco 44270, Mexico; gdedios_al@ciatej.edu.mx (G.T.d.D.-F.); jaaguilera_al@ciatej.edu.mx (J.d.R.A.-M.); 2CONACYT-Department of Medical and Pharmaceutical Biotechnology, Centro de Investigación y Asistencia en Tecnología y Diseño del Estado de Jalisco A.C. (CIATEJ), Av. Normalistas 800, Colinas de las Normal, Guadalajara, Jalisco 44270, Mexico; tcamacho@ciatej.mx

**Keywords:** 3D cell culture models, SARS-CoV-2, organoids, antiviral therapeutic drug discovery

## Abstract

In the last decades, emerging viruses have become a worldwide concern. The fast and extensive spread of the disease caused by SARS-CoV-2 (COVID-19) has impacted the economy and human activity worldwide, highlighting the human vulnerability to infectious diseases and the need to develop and optimize technologies to tackle them. The three-dimensional (3D) cell culture models emulate major tissue characteristics such as the in vivo virus–host interactions. These systems may help to generate a quick response to confront new viruses, establish a reliable evaluation of the pathophysiology, and contribute to therapeutic drug evaluation in pandemic situations such as the one that humanity is living through today. This review describes different types of 3D cell culture models, such as spheroids, scaffolds, organoids, and organs-on-a-chip, that are used in virus research, including those used to understand the new severe acute respiratory syndrome-coronavirus 2 (SARS-CoV-2).

## 1. Introduction

Viral infections have become a global public health problem [1,2]. In the last two decades, severe acute respiratory syndrome-coronavirus (SARS-CoV) in 2002, Middle East respiratory syndrome-coronavirus (MERS-CoV) in 2012, and recently, the new coronavirus SARS-CoV-2 in December of 2019 [1] were the causes of significant outbreaks [3]. COVID-19 has spread globally and brought more than two million deaths [1]. This situation has impacted the economy and human activity worldwide [4]. Other outbreaks, such as Ebola in 2014 and AH1N1 in 2009, have also caused thousands of deaths worldwide [1,5]. The Zika virus outbreak in 2016 increased the number of pregnancy complications and Brazilian children born with microcephaly [1]. Other infectious diseases will continue to emerge in the future [6]; thus, improving pre-clinical strategies to develop new therapeutic compounds or vaccines to overcome public health emergencies is indispensable.

When a new virus outbreak is detected, it is necessary to quickly and reliably elucidate the infection mechanism and the virus–host interactions [7]. The preliminary conclusion is that the illness probably cannot be treated with previously approved conventional antiviral therapies [2]. Furthermore, many emerging viruses spread more rapidly than we can act, resulting in high fatality rates. Current laboratory discoveries are primarily based on monolayer cell cultures (2D cultures) and animal models to assess pre-clinical stages of new therapeutic drugs or repurpose existing pharmacological compounds. Since the emergence of SARS-CoV-2, the international scientific community has analyzed and started describing the infective mechanisms of this pandemic virus to bring effective therapeutic drugs and vaccines to confront COVID-19. Because of this, testing possible antiviral drugs and accelerated vaccine development must be a priority for the scientific community, and we need reliable testing models to achieve this goal.

The 3D cell culture pathophysiological models could help, and provide preliminary approaches or replace tests in animal models that are expensive and time-consuming in the preclinical stage of new therapeutics. The 3D cell culture pathophysiological models could overcome some of the weaknesses of monolayer cultures and provide preliminary approaches before animal testing to accelerate the research in drug/vaccine development for newly emerging viruses. Here, we review 3D cell cultures models used to investigate viral infections, including those used to research the new coronavirus SARS-CoV-2.

## 2. Classical Approach: 2D Culture and Animal Models

### 2.1. Monolayer Cell Culture

In monolayer cell cultures, cells grow on polystyrene plates. These in vitro models have been used to study viral infection. Monolayer cell cultures control many experimental parameters, and possess apparent advantages such as being more straightforward, less time-consuming, and cheaper than in vivo experiments. An inconvenience is that monolayer cell cultures usually use a single cell line, making it impossible to evaluate the interaction between different cell types and resemble the organization and heterogeneity of tissues [8,9]. When cells have been grown attached to a plastic surface, they have probably lost the phenotype, polarity, and tight junctions present in the target tissue. These changes lead to antiviral drugs’ efficacy misestimations [10,11]. Additionally, some viruses cannot replicate in monolayer cell cultures. In these cases, the study of their pathogenesis and replication requires models that reproduce the affected tissue environment [12].

Stem cells have been cultured in 2D and differentiated into specialized cells to study viruses. These models have contributed to understanding different diseases, but lack the tridimensional microenvironments relevant to stem cell final differentiation and in vivo regulation [13,14]. Human induced pluripotent stem cell (HiPSC) 2D models have been used to study the tropism of neurotropic viruses. However, this type of model cannot mimic the blood–brain barrier (BBB), failing to recapitulate the virus’s entry into the central nervous system. It is possible to imitate the complex BBB functions using organoids and microfluidics devices [15,16]. The 3D models can emulate characteristics, in vivo responses, organ, and tissue functions that are impossible to generate in conventional methods.

### 2.2. Animal Models

Animal models are still considered essential in human disease studies as conventional study models because they provide information regarding the interaction between several organs and viral infections. They have a shorter life span, making it possible to study diseases during the complete animal life cycle; in addition, animal testing is a requirement in preclinical testing phases [17]. Animal model selection depends on the research objectives, such as vaccine, pathogenesis, or antiviral drug evaluation. For vaccine research, the animal model must demonstrate immune response differences between vaccinated and placebo groups, besides viral load diminution in tissues and neutralizing antibody-increased titers. If the objective is pathogenesis elucidation, the animal model should replicate the interactions between the immune system and viral infection. For evaluation of antiviral drugs, animal models should express viral receptors like in human tissue, and mimic the original viral infection route [18].

Nevertheless, animal models show some limitations, that is, they cannot reproduce human virus pathophysiology due to human cell tropism. These disadvantages are associated with insusceptibility to human respiratory viruses [19,20] or required high-dose inoculums that hinder understanding of the pathogenesis [21]. Small animals can be genetically modified, such as chimeric [22] and transgenic [23] animal models. However, even with transgenic animal models, it is difficult to translate the results to human disease due to differences in the immune system between species [24]. The pathophysiological response can significantly differ from the human response and mislead the experimental results [25].

Additionally, ethical concerns regarding the use of animals are another consideration: first, using experimental animals should be avoided when possible; secondly, the number of animals used per experiment should be reduced, and finally, methods to minimize animal suffering should be implemented [26]. Furthermore, the animal models must provide consistent and reproducible results. All these disadvantages may restrict the attempt to obtain reliable experimental preclinical models.

## 3. Generalities in 3D Cell Culture

Three-dimensional cell culture attempts to mimic the spatial organization of tissues. The 3D culture systems can be divided into scaffold-based, scaffold-free, and hybrids [27,28]. The scaffold-based systems use structures that mimic the extracellular matrix (ECM) composition to simulate the native cellular microenvironment. In scaffold-free systems, cells aggregate and self-assemble, as occurs in natural processes of organogenesis [29]. The hybrid systems include scaffold-base characteristics like synthetic matrix and external physical supports that confer more complex interactions between cells and ECM-cells. A clear example of these models is the organotypic raft culture and organ-on-a-chip systems incorporating microfluidic dynamics and gas exchange into simulated atmospheres with permeable membranes (Figure 1). In all 3D cell culture systems, the cells grow in a context that regulates their phenotype and function, not just as an isolated entity [30], making 3D cell culture systems more representative of what occurs in vivo compared with monolayer cultures.

In 3D cell cultures, it is possible to simulate native cellular microenvironment, cell–cell interaction, cell–ECM interaction, and cell polarity, and to test drugs with diffusion gradients, or tight junction barriers [31]. These capabilities make them a good model for applications such as cell biology research and disease pathophysiology, and a valuable tool for drug evaluations and in vitro toxicological studies [32,33].

Summarizing the advantages and disadvantages of 2D cell culture, animal, and 3D cell culture models (Table 1) as previously described, the analysis reflects the need to optimize and, in some cases, modify the strategies to elucidate virus–host interactions.

### 3.1. Virus Infection Research in 3D Cell Models

The general infection process of a virus includes multiple interactions with the host cell [34]. Viruses attach to a host cell tight junction’s barriers surface receptor, enter the cytosol, transport to the replication site, and use host cellular machinery to replicate. Whether the infection process results in symptomatic or asymptomatic patients depends on the host cell behavior and the infected tissue environment. These kinds of host–pathogen interactions are rarely studied and require an integrative approach, including 3D cultures as experimental ex vivo and in vitro results that provide information on pathogen diffusion, infectivity and replication dynamics, live imaging of cell motility, and cell–cell interactions [9], as well as a computational approach to the analysis of intricate images, and at last, mathematical modeling to incorporate an independent process in dynamics depicting the natural viral infection [35]. Moreover, 3D cell culture microenvironment simulation offers a good model for the study of virus–host cell interaction, the viral infective cycle, oncogenic viruses, new anti-viral drugs, and behavior used to study pathogens that are not cultivable in 2D models.

### 3.2. Spheroids

Spheroids are the 3D cell culture model most used in biomedical research [36]. The use of multi-cellular tumor spheroids started in the 1970s as an in vitro model of tumor cell response to therapies [37]. They can be generated by diverse techniques, like using non-adhesive surfaces, or forces like gravity, centrifugation, constant stirring, electric fields, magnetic force, or ultrasound [38]. Spheroids show some characteristics similar to those observed in the tumor microenvironment, that is, hypoxia and tumoral cell-to-cell interaction. These characteristics provide a valid model to study the altered metabolic pathways in cells infected by viruses associated with human cancers, that is, hepatitis B, hepatitis C, papilloma, Epstein-Barr, herpes, and Merkel cell polyomavirus [39]. Choi et al. used a spheroid model and demonstrated that the metabolism of cells infected by human herpesvirus-8 (HHV-8) was different in 2D than in 3D. They concluded that the interaction of a virus oncoprotein with a host enzyme in spheroids increased intracellular amino acid concentration and promoted 3D cell growth. In in vivo tumorigenesis, this amino acid metabolic pathway could become a target for cancer therapy [40].

The cells in the center of spheroids receive fewer substrates, that is, oxygen and glucose, than those in the periphery, leading to necrosis [31]; this is similar to the poorly vascularized solid tumors, where the delivery of oxygen and nutrients is limited regarding the cells closer to the blood vessels. The necrotic core might affect the efficacy of treatments due to limited drug access or their activity being affected in a hypoxic, acidic, or nutrient-deprived microenvironment [41]. Spheroids are ideal models for in vitro drug testing since they allow the analysis of drug effects in the context of diffusion gradients, adhesion, and tight junction barriers [31]. They are also valuable for altered gene expression studies and evaluating liposomes, nanoparticles, and antibody-based therapies [42].

Saleh et al. used alveolar type II epithelial cells’ spheroids to study respiratory syncytial virus (RSV) infections. The spheroids present syncytia formation and mucin overexpression, similar to lung epithelial tissue during infection. The syncytia formation is not present in alveolar epithelial cell lines grown in monolayer culture [43].

The spheroids demonstrated the advantage of self-organization and the formation of cortical layer-like architecture properties [44,45]. Bortolotti et al. used a spheroid model to evaluate the effect of human herpesvirus 6 (HHV-6A) on microglial cell status to analyze the genomic expression in Alzheimer’s pathogenesis [46].

In a hybrid system, using a matrix or a scaffold to support the spheroids can recapitulate in vivo cell polarization and microenvironment features, providing better insights into drug responses. This cell culture technique has widely been used in cancer research and drug testing [47]. The hybrid systems can also be relevant in viral infection research since cell polarity can influence viral receptor expression. The localization of these receptors is an essential key in the infection process and virus tropism [48]. Ananthanarayanan et al. used a galactosylated cellulosic sponge to form spheroids of a co-culture of Huh 7.5 cells infected with hepatitis C and primary human hepatocytes. They developed a method to obtain spheroids in which cells maintained their basolateral and apical domains and expressed all hepatitis C virus entry receptors. This system would serve to study events such as chronic toxicity and progression of hepatotropic infections, and it can be applied to screen antiviral drugs [49]. Hybrid systems have the advantage of emulating more complex microenvironments, and provide information on both scaffold-based and scaffold-free models.

Unlike 2D cell culture methods, spheroids can replicate some of the hallmarks observed using in vivo models. This property can help in the discovery of new therapeutic targets and virus-host interactions. One essential disadvantage of this model is that the formation of spheroids can be a complicated and labor-intensive process, leading to a high degree of variability that hinders experiment standardization associated with the homogenous spheroid size.

### 3.3. Scaffold Base

The use of scaffolds can overcome some of the 2D cell culture limitations; they mimic the ECM composition or physical properties to simulate the native cellular microenvironment [30]. Several biomaterials and methods can be used to generate structures that support cell growth and provide physical and biochemical stimuli for optimal cell organization and differentiation. They can be adapted to the mechanical and physical characteristics required to study tissue physiology and pathophysiology [50,51].

Scaffolds and matrixes have diverse applications in 3D cell culture and regenerative medicine, but they also have some limitations. Scaffolds can simulate cell–ECM and cell–cell interactions; therefore, these systems can elucidate ECM and tight cell junctions that influence the cytotoxic effect of compounds and treatments [52,53]. Changes in physical signals from tissue microenvironments can alter gene expression and cell behavior [54]. The softness or stiffness of matrices, such as hydrogels, can be controlled by varying the concentration and nature of their components or controlling the crosslinking process. These matrices are suitable to carry out mechanotransduction studies [55]. They are also ideal for cell behavior studies, delivery applications, toxicology studies, drug discovery, and biomaterials testing, as well as their application in tissue engineering. Despite their multiple applications, there are some challenges with the use of scaffolds and matrices. The biopolymers used can vary between batches, affecting reproducibility [56]; additionally, the design and polymerization strategy impact biodegradability and pore size. Both characteristics are critical for the scaffold’s biocompatibility [57], and it is challenging to extract cells from the matrices for further analysis.

The matrix and scaffold base methods permit evaluating ECM components’ possible effects on the virus–host cell interaction. In 2003, Thach and Stenger used baby hamster kidney (BHK) cells in type I collagen culture to demonstrate that the Sindbis virus could penetrate the gel matrix and infect cells; the virion did not interact significantly with the collagen matrix [58]. A recent study used a collagen 3D scaffold and computational analyses to test the influence of environment and CD4+ T cells’ motility on human immunodeficiency virus (HIV) spread. This study determined that the physical properties of the ECM could limit virions diffusion and suggested that HIV-1’s spread depends on cell-associated transmission [9].

The use of matrixes and scaffolds enables gathering information about the integration of the cells with their environment. This feature is essential in virology studies due to the need for virions to diffuse in the extracellular medium before they attach to cells [59]. The combination of scaffold systems, bioinformatics, and imaging technologies brings deep insight into virus infection processes and cell function mechanisms.

### 3.4. Organotypic Raft Culture

Three-dimensional organotypic rafts cell culture systems consist of epithelial cells, that is, keratinocytes seeded on the top of a dermal equivalent matrix made of fibroblast and collagen type I. These systems include an air–liquid interface that permits the complete differentiation of keratinocytes [60] obtained from immortalized cell lines [61] or directly from patient donors [62].

Organotypic raft cultures accurately mimic the morphology and physiological features of the epithelium [63]; thus, they are the primary model used in epithelial differentiation and epithelial cancers research [64]. In addition, they are essential in the human papillomaviruses (HPV) replicative cycle research [65]. Even with their simplicity, it was possible to elucidate the contribution of the interactions between the stroma and the epithelial compartment to HPV genome maintenance and establish the viral oncogenes E6 and E7’s invasive potential ability to alter the secretory profile of epithelial cells [66].

This 3D cell culture system can also be used to study the replication, latency, or persistence of other viruses that target the epithelium, such as herpes simplex viruses (HSV), varicella-zoster virus (VZV) [67], poxviruses, adenoviruses, and parvoviruses [60], and to evaluate the activity of antiviral agents [68,69,70].

### 3.5. Organoids

Organoids are a three-dimensional culture derived from human stem cells and patient-derived induced pluripotent stem cells that mimic some specific organ functions [71]. Organoid technology applies in multiple infectious diseases, monogenic hereditary diseases, and personal and regenerative medicine [72].

Organoids have been used to study intestinal, neuronal, and respiratory viral infections [19] and have been helpful to investigate pathogens not characterized before [73]. In 2012, stem-cell-derived intestinal organoids served as a model to study rotavirus replication, establishing the potential of this cell culture model in virus biology research [74].

Since this achievement, organoids have been more regularly used to investigate viruses [75]. Enteroids, a kind of organoids derived from intestinal stem cells, have been used to understand the human’s intestinal physiology and the pathophysiology of gastrointestinal infections [76]. Liver organoids generated from human induced pluripotent stem cells were used to recapitulate hepatitis B virus (HBV) infection, and mimic virus-induced hepatic dysfunction [77]. It has also been possible to develop organoids to model diseases from complex organs, like the human brain [78]. Brain organoids allow the analysis of Zika virus infection, and their effect on cerebral architecture and function [79]. The promising models based on organoids could change how development biology and pathophysiology are studied now [78]. The future challenges of the organoids-based models center on the lack of vasculature and immune cells, and also the costs and time involved. Another challenge to address is the variability between experiments, focusing on the heterogeneity of these models. Once these problems are solved, the organoids could be in vitro gold standards in numerous diseases [80,81].

### 3.6. Organ-on-a-Chip

Organs-on-a-chip is a microfluidic device in which cells are seeded and perfused in a chip-like array. They aim to recapitulate the minimal functional units of an organ or a tissue [82]. Microfluidic devices allow the control of microscale flows, making it possible to simulate the human body’s blood circulation pattern. In this way, it is possible to connect multiple organs-on-a-chip by channels to reproduce the interactions between organs. These bodies-on-chips can account for multiorgan interaction in drug pharmacokinetics studies and toxicity tests [83].

Viral infection research has included liver chips, gut chips, nervous systems chips, kidney chips, and lung chips [84]. Ortega-Prieto et al. used a liver-on-a-chip composed of primary human hepatocytes with constant medium perfusion to study HBV infection. The study moved forward with a liver-on-a-chip designed for long-term outcomes of HBV and other hepatotropic pathogens and drug tests [85].

A 3D-printed system emulating the glial cell-axon interface was developed by Johnson et al. to understand the transport and infection process of pseudorabies virus (PrV) in the nervous system. The printed system consisted of three chambers, each with a different cell line type (Figure 2A). They infected chamber number two and observed the axon-to-cell spread of pseudorabies viral particles; the results suggested a bottleneck to virus transmission from superior cervical ganglia neurons axons to hippocampal neurons and Schwann cells [86]. Using a distal-renal-tubule-on-a-chip composed of distal renal epithelial cells in a three-layered format microfluidic chip, researchers showed that the pseudorabies virus (PrV) could induce renal dysfunction in electrolyte regulation. This chip made it possible to imitate the distal renal barrier structure and sodium reabsorption function. The authors found that sodium reabsorption decreased in PrV-infected systems and showed that the microfluidic chip is suitable in virus pathogenesis research [87].

Nawroth et al. used a lung-on-a-chip that recapitulated the mucociliary airway epithelium to explore human rhinovirus (HRV) infections in healthy and asthmatic conditions. This microfluidic chip was previously coated with IL-13 to induce a lymphocyte Th2 microenvironment that imitated the cytopathology’s key hallmarks and human inflammatory responses [88]. A lung-on-a-chip can also be used to investigate viral-bacterial co-infections. Deinhardt-Emmer et al. studied the spread of *Staphylococcus aureus* in a co-infection setting with influenza virus using an alveolus model (Figure 2B). This system emulated some of the pneumonia features. The results suggested that a co-infection of influenza virus and *S. aureus* affects the endothelial barrier’s integrity [89].

Organs-on-a-chip can help elucidate host–pathogen interactions, carry out analysis of infection processes, evaluate the interaction of different cell types and interactions between various organs, and summarize some of the disease’s pathophysiological aspects. They also provide the possibility to create a dynamic and controllable microenvironment with continuous nutrition delivery and a waste removal system. As with other 3D systems, the organs-on-a-chips exhibit variation between different manufacturing batches; furthermore, it can be complex to carry out high-throughput studies [90]. The organs-on-a-chip models focus on some aspects of the immune system, like cell migration, intended to reproduce the dynamic and complex interactions in cells that produce chemokines and the cytokines from cell co-cultures. The immune responses following from infections and the adaptive immune reactions allow the evaluation of how immune cells react and decide to migrate in response to cytokines in real-time [91]. There is an advantage over other 3D cell cultures, which brings the organs-on-a-chip systems to the forefront for potential use as preclinical platforms to develop new drugs and vaccines.

## 4. Three-Dimensional (3D) Cell Culture in SARS-CoV-2

The COVID-19 disease caused by the SARS-CoV-2 coronavirus has led to an increase in hospitalizations for pneumonia. COVID-19 was first reported in December 2019, when an outbreak of unidentified pneumonia emerged in Wuhan, China [92]; since then, the disease has been propagated rapidly by human-to-human contact [93]. In March 2020, the World Health Organization (WHO) declared it a pandemical event. A year after the pandemic start, the WHO has reported more than 121 million infections worldwide.

SARS-CoV-2 is a single-stranded, positive-sense ribonucleic acid (+ssRNA) of ~30 kb in size, enveloped by a lipid membrane. The viral genome codifies four structural proteins: spike glycoprotein (S), small envelope glycoprotein (E), membrane glycoprotein (M), nucleocapsid protein (N), and sixteen nonstructural proteins (Nsp1-16) implicated in viral replication and pathogenesis, and another nine accessory factors [94,95]. SARS-CoV-2 uses the angiotensin-converting enzyme 2 (ACE2) as a critical receptor for infection through S glycoprotein’s binding, allowing the entry of the virus through endocytosis (Figure 3) [96]. The ACE2 receptor is expressed in such diverse organs and tissues as the heart, blood vessels, kidneys, intestine, and lungs; it also has a vital role in the cardiovascular and immune systems [97]. The diverse localization of the ACE2 receptor and the impact of the virus infection have focused the current therapeutical effort on avoiding the viral protein S interaction with receptor ACE2 using antibodies or soluble proteins [98].

SARS-CoV-2 infects the respiratory tract, spreads throughout the body, and infect various organs depending on the severity of the disease. Clinically, the infection can be asymptomatic, and it can manifest with mild symptoms, multi-organ failure, or death [99]. Several studies have associated old age, cardiovascular disease, diabetes, chronic respiratory diseases, hypertension, and cancer with an increased risk of death. However, Jordan et al. report inconclusive data from the analysis of how the risks associated with underlying co-morbidities can vary in different population groups or settings [100].

The transmission of the virus is associated with contact among infected people through respiratory droplets or contaminated surfaces [101]. Due to the rapid spread, mortality rate, and poor understanding of the long-term effects of SARS-CoV-2 infection, the scientific community has employed various tools to understand the mechanisms involved in the pathogenicity, transmission, and treatments of COVID-19. One tool that has become very important to elucidate aspects of the infection’s biology are the 3D culture systems, which stand as a new approach to evaluate cell tropism and damage to different tissues (Figure 3), and help to develop new anti-viral therapies and vaccines.

Several cell lines have been used in 3D cultures to gather important information concerning cell biology of viral infections and drug evaluations (Table 2), making them suitable for SARS-CoV-2 viral studies. The most widely used models are based on stem cell organoids. According to the organ under investigation (Figure 4), they can be classified as pulmonary or airway models, and extrapulmonary models such as intestine, heart, and brain.

### 4.1. Pulmonary Models

Pulmonary models derived from induced pluripotent stem cells (iPSC) and embryonic stem cells (ESC) can emulate host characteristics and function as tools to evaluate aspects regarding viral replication, tissue tropism, and immune response [116,117]. SARS-CoV-2 infection models using organoids generated by Han et al. have provided answers on cellular metabolism in SARS-CoV-2 infection, possible alternative routes of viral entry, and viral spreading that can potentially be used in high-throughput screenings of FDA-approved therapeutic candidates [108,118].

Alveolar models of the proximal-distal axis developed as organotypic raft cultures were essential to determine the infection process of cells from the proximal respiratory tract by SARS-CoV-2. The infection process was quick and targeted both ciliated and goblet cells. In the distal axis model, activated HAT2 cells induced a series of pro-inflammatory transcripts, such as interferon 1 (IFN-*β*1) and interferon 3 (IFN-3). Several drugs were evaluated, such as beta interferon (IFN-*β*1), hydroxychloroquine, and remdesivir. It was shown that, unlike hydroxychloroquine, remdesivir inhibited viral replication at a greater rate than IFN-*β*1 independently of the epithelium’s origin [105]. This model showed that immune cells play an essential role in alveolar barrier dysfunction due to the expression of cytokines such as interleukin 6 (IL-6) and interleukin 8 (IL-8) induced by a viral infection, which damaged the alveolar barrier.

These observations correlate to the findings in the postmortem lung tissues of patients severely affected with COVID-19, pointing to a possible relationship with clinical manifestations such as severe tissue damage, thromboembolism, and excessive inflammation [119]. Remdesivir inhibited viral replication in the model, leading to restoration of the damage to the epithelial and the endothelial layers. The alveolar infection model can be used to explore the effects of remdesivir administration combined with other drugs [114].

### 4.2. Extrapulmonary Models

Some extrapulmonary infection models have been used to study the neuropathology associated with COVID-19. Through organoids, Ramani et al. found that SARS-CoV-2 targets human neurons despite the low expression of ACE2 [109]. The use of extrapulmonary models opens the possibility of studying another viral entry point, that is, endolysosomal pathway mediated by cathepsin-L in cardiomyocytes [120]. Buzhdygan et al. reported that the SARS-CoV-2 spike protein-induced destabilization of the blood–brain barrier, promoted a pro-inflammatory state but did not acutely alter cell viability. In addition to that, they show that SARS-CoV-2 can induce micro cloth formation in the vasculature of peripheral tissues and within the central neural system (CNS) [104,121]

Clinical evidence suggests that the gut is another high-risk organ for SARS-CoV-2 infection [122]. Through intestinal organoids and organs-on-a-chip, it was possible to observe the active replication of SARS-CoV-2 in some types of cells present in the gastrointestinal tissues, suggesting that they could be target cells of SARS-CoV-2. It is essential to highlight that not all cells express identical amounts of ACE2. These findings suggest that infection could disrupt the absorption, metabolite transport, hamper hormonal secretion, and affect local immune defense [92,123]. It was also possible to assess the intestinal barrier, showing that the injury caused by the virus could lead to diarrhea, among many other gastrointestinal symptoms frequently reported in patients with COVID-19 [115]. In the intestinal organoids developed by Krüger et al., the effect of drugs such as remdesivir, famotidine, and the EK1 fusion peptide showed a concentration-dependent action. Unlike famotidine, EK1 and remdesivir inhibited virus replication without cytotoxic effects at higher concentrations [123].

In addition to respiratory complications, cardiovascular diseases have emerged as an essential indicator of poor prognosis in COVID-19. Cardiac injuries present in non-severe patients could translate to long-term cardiac pathologies after the infection is solved [97]. Bailey et al. observed using a cardiac tissue model that SARS-CoV-2 selectively infects cardiomyocytes. They stated that viral entry depends on ACE2 and an endosomal cysteine protease activity, highlighting the possibility that alternative mechanisms may facilitate the SARS-CoV-2 viral access into the human heart [124]. Perez et al. showed that SARS-CoV-2 infects cardiomyocytes through an endolysosomal pathway, specifically through the use of the cathepsin-L protease [120].

Acute kidney injury is one of the most frequent organs damaged in patients with severe COVID-19. It is well known that ACE2 is highly expressed in proximal tubule cells [125]. Monteli et al. developed kidney organoids containing groups of cells that expressed ACE2, like observed in native tissue, producing infectious virus progeny that could be inhibited by soluble ACE2 recombinant human (hrsACE2) protein. This finding suggests that hrsACE2 could protect the lung from injury and block the entry of SARS-CoV-2 into its target cells [110]. However, it is unclear whether a cytopathic effect induced by coronavirus infection is responsible for injuries caused by immunopathogenic damage.

A proportion of COVID-19 patients have reported liver dysfunction. In a retrospective study conducted in Shanghai, Fan et al. observed that around 50% of patients between 15 and 88 years with the disease showed liver dysfunction, particularly those with severe disease. That liver damage could be related to the direct cytopathic effect of the virus, uncontrolled immune reaction, sepsis, or drug-induced liver injury [126]. Zhao et al. developed organoids of the hepatic duct evaluating the cellular tropism of SARS-CoV-2 in infected cholangiocytes, yielding information on the pathophysiology. The infection induces cholangiocyte cell death and alters the barrier functions of bile acids, which could explain the liver damage observed in patients with COVID-19, due to the accumulation of acid biliary [113].

In addition, 3D models have brought great insights into the pathophysiology of infection, accelerating the understanding of virus–host interactions, and developing more specific therapeutics to establish treatments during an acute episode of COVID-19 or even post-COVID-19 (i.e., pulmonary fibrosis) [127]. These findings contribute to identifying diagnostic and therapeutic objectives more quickly and efficiently. Although the 3D technologies apply to biomedical investigations, and many advances apply to basic research, the pandemic’s events have forced us to undertake the endeavor of speeding up its clinical and translational relevance, making robust 3D models, and using standardized procedures to mimic human pathophysiological processes, projecting the eventual replacement of animal models. As an example of 3D cell culture contributions to anti-viral drugs against COVID-19, the organotypic raft culture tool was used to evaluate remdesivir, approved by the Food and Drug Administration (FDA) for COVID-19 treatment in hospitalized adult and pediatric patients. Remdesivir shows favorable patient outcomes and demonstrates the translational potential of 3D technologies not just in pandemic events, but also in more favorable times.

Despite the clinical and postmortem evidence seen in patients with COVID-19, this illness can be classified as a vascular disease associated with disruption of the immune, renin-angiotensin-aldosterone (RAA), and thrombotic balance, all of which converge on the vascular endothelium as a standard pathway [128]. At the moment of writing this review, there are a few 3D models reported (see Table 1) of COVID-19 that recognize the vascular-endothelial tissue as an axis of organic damage. The description of the pathophysiology with this new perspective implies refined and accurate vascular 3D cell culture models, which positively impact the research focused on new therapeutic drugs discovery and vaccines to treat this illness.

## 5. Conclusions

The COVID-19 pandemic challenged the scientific community’s capacity to research, develop, validate, and approve vaccines and anti-viral drugs in a short period. Here, we reviewed 3D cell culture models used in SARS-CoV-2 research. The most-used model to study this new virus is the organoid system. Its main application was to evaluate virus tropism in the lung, heart, pancreas, liver, and neural organoids. To the best of our knowledge, only three studies reported the use of 3D models for drug-testing purposes (a spheroid, an organo-typic raft culture, and an organ-on-a-chip). Even though monolayer cell culture models can underestimate the anti-viral efficacy, they are still extensively used to evaluate potential drugs to treat COVID-19, probably because of their simplicity, low cost, and reproducibility.

In the race to get efficient therapeutic drugs and effective vaccines, we need to consider the proper usage and reaches of the monolayer culture approach, 3D cell cultures, and animal models to obtain reliable results and interpretations that represent the natural viral replication and infective cycle in humans. Research models should be used according to the specific purpose and when their use is justified; nevertheless, there is a need to develop more effective research models. Applying tridimensional models in the early stages of the drug development process may contribute to anti-viral drugs or vaccine discovery, as 3D cell cultures can resemble complex microenvironments, cell–cell interactions, and in the case of organ-on-a-chip systems, can even emulate interactions with immune cells. Nevertheless, aspects such as variability, experiment standardization, and lack of vascularization in the models need to be evaluated and improved to make them more robust with adequate tools, that is, effective optical microscopy, computational data analysis, and mathematical modeling to analyze and interpret results derived from these in vitro 3D experiments [127]. A recent publication states that COVID-19 is more a vascular endothelial disease than a purely pulmonary disease [128]. In this sense, 3D cell culture variants need to refine their deficiencies in vascular-endothelial representation to efficiently describe the pathogenesis and possible implications in the short and long term in heart conditions, BBB alterations, pulmonary vasculature, or placenta circulation as well.

Current technologies such as omics, artificial intelligence, and bioinformatics databases can infer, analyze, and predict data. These technologies strengthen the 3D cell culture approach and, consequently, accelerate the acceptance in preclinical stages of new anti-viral therapeutics and vaccines. They will shortly replace animal experiments, which is a critical step toward reaching clinical trials of those developments.

In the actual sanitary global emergency context, 3D cell culture models attempt to provide a different approach, closer to human pathophysiology than the conventional in vitro, even in vivo, animal models. In this way, research and development associated with these technologies should be encouraged in a global effort to combat the viral infection that has become a humanitarian crisis.

## Figures and Tables

**Figure 1 biomedicines-09-00602-f001:**
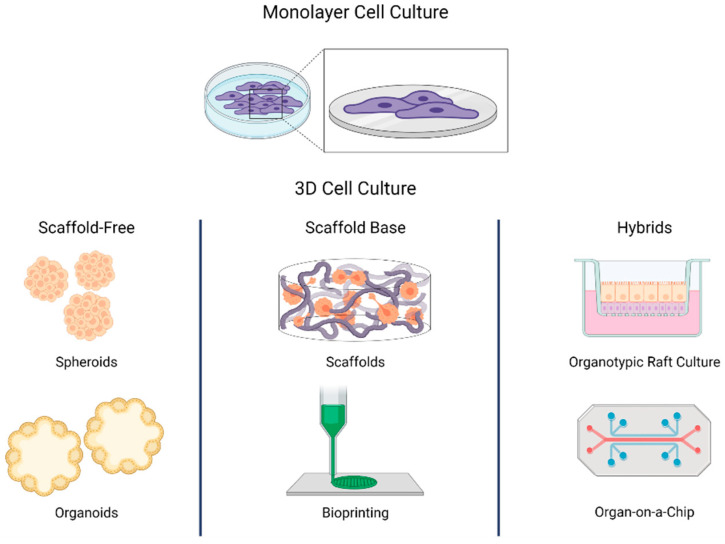
Scheme of diverse 3D cell culture strategies (i.e., scaffold-free, scaffold-base, and hybrids). In monolayer cell culture, cells grow attached to a plastic base. In 3D cell culture, cells self-assemble or grow in structures that resemble the extracellular matrix. The 3D cell culture can be divided into three groups: scaffold-free, scaffold base, and hybrids. In scaffold-free systems, cells aggregate as occurs in natural processes of organogenesis. Scaffold-based systems use structures that mimic the extracellular matrix. Hybrids use a matrix or a scaffold to support scaffold-free systems.

**Figure 2 biomedicines-09-00602-f002:**
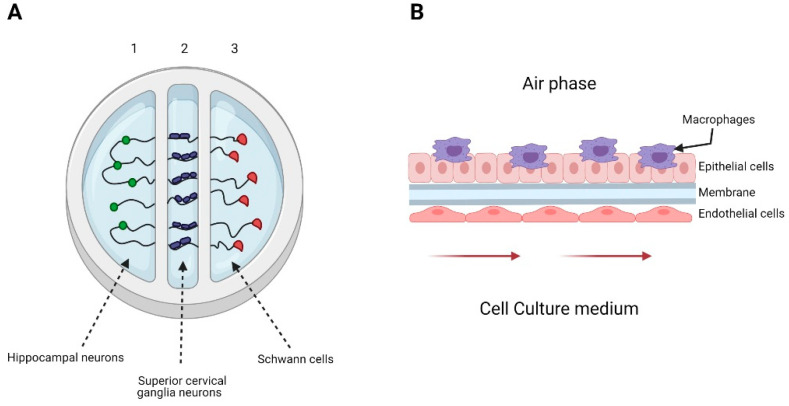
Examples of organs-on-a-chip used in viral infection studies. (**A**) Representation of the 3D-printed system developed by Johnson et al. The system consists of 3 chambers, seeded with hippocampal neurons in chamber 1, superior cervical ganglia neurons in chamber 2, and Schwann cells in chamber 3. They only infected chamber 2 with pseudorabies virus. (**B**) Representation of the alveolus model generated by Deinhardt-Emmer et al. The system consisted of a chamber divided by a porous membrane. The upper phase was seeded with a co-culture of alveolar epithelial cell and monocyte-derived macrophages and exposed to an air phase. The lower side contained endothelial cells, and the culture medium was perfused using a peristaltic pump.

**Figure 3 biomedicines-09-00602-f003:**
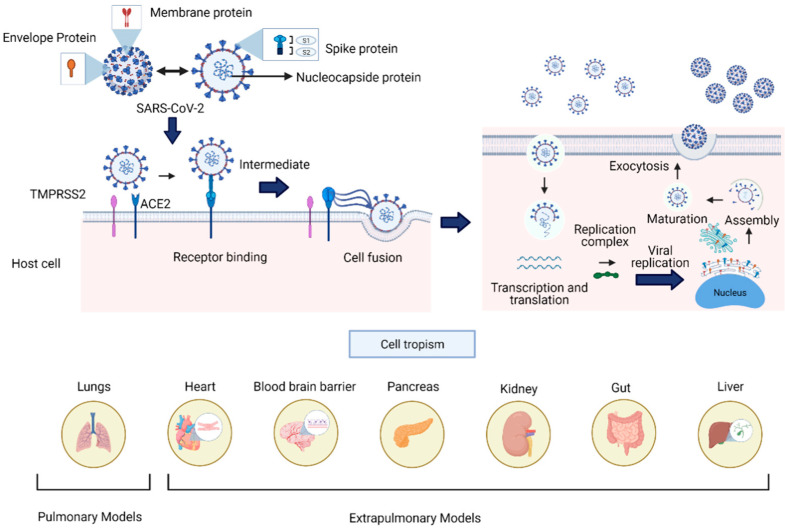
SARS-CoV-2 viral structure, infective cycle, and cell tropism. The structural proteins of SARS-CoV-2 comprise the S (spike), N (nucleocapsid), M (membrane), and E (envelop) proteins. For viral entry into the host cell, the spike protein interacts with the ACE2 and TMPRRS2 enzymes widely distributed in several human tissues. The infective cycle of SARS-CoV2 includes endocytosis mediated by the viral spike protein–ACE2 cell interaction. Once entering the cell, replicating and forming virions is achieved by the assembly and maturation of new viral particles and subsequent exocytosis. The lower panel shows the target organs previously developed to simulate viral infection processes. These 3D cell cultures resemble in vitro aspects of human pathophysiology, which are crucial to understand and address the current pandemic.

**Figure 4 biomedicines-09-00602-f004:**
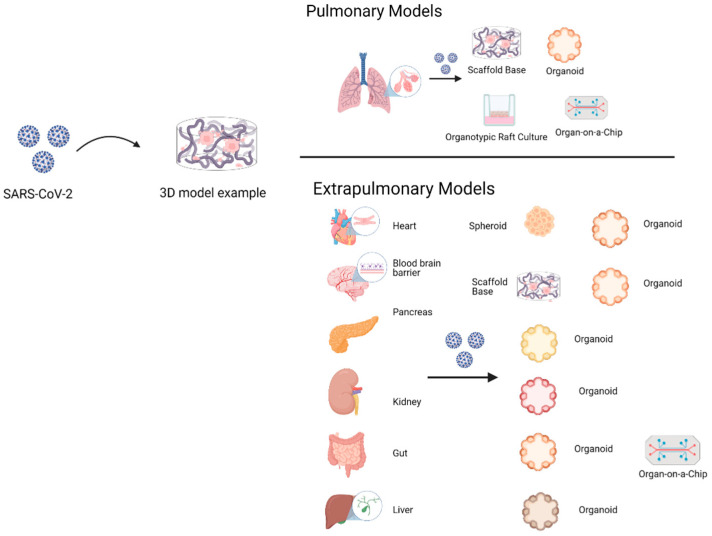
The 3D cell culture pulmonary models include scaffold-base, organoids, organotypic raft culture, and organ-on-a-chip. The objective is to emulate viral tropism, infective cycle, and replication process. Instead, 3D cell culture extrapulmonary models have been evaluated in organoids and organ-on-a-chip, except for the blood–brain barrier, which only includes scaffold-base models. The gut-on-a-chip model includes a few of the 3D cell culture models where therapeutic drugs like remdesivir are evaluated.

**Table 1 biomedicines-09-00602-t001:** Comparison of 2D and 3D cell cultures and animal models in viral infection assays.

Characteristic	2D Cell Culture Models	Animal Models	3D Cell Culture Models
Spatial distribution and cells–cell interaction	When cells grow attached to a plastic surface, they probably lose the phenotype, polarity, and tight junctions naturally present in the target tissue. This leads to modifications in the viral infective cycle.	Can differ from human cells in the presence, number, and distribution of viral ligands, thus affecting the replicative and infective viral cycle.	Confers a three-dimensional organization closer to human tissue, providing the cells with apical-basal polarity and cellular interactions, resembling in vivo microenvironments proper for virus–host interactions.
Virus–host interactions	Their simplicity allows controlling of most of the experimental variables. However, with these models it is impossible to evaluate the immune reaction and the viral infective cycles.	Murine models are the most used on in vivo assays; nevertheless, they cannot imitate the human pathophysiology in viral infections because of the viral tropism to human cells or the higher concentration of viral inoculum needed to evoke disease.	They can simulate native cell–cell communication and cell–ECM interaction. The organ-on-a-chip systems allow the creation of dynamic and controllable microenvironments proper for viral infections and immune response analysis.
Reproducibility	These systems are well-characterized assays, with high reproducibility and with the availability of a detailed bibliography for consultation and comparison between viral infection models.	The reproducibility of preclinical research involving animal models with some respiratory viruses is inaccurate, particularly associated with the viral infection cycle.	Depending on the 3D model selected, the fabrication can be labor intensive and time consuming. Nevertheless, the organotypic raft cell culture is suitable for high-throughput screening.
Vasculo-endothelial emulation	They are usually monocultures that only allow the study in a single cell type. The vasculature complexity is not convenient to replicate with this approach.	As complete organisms, animal models are essential in pharmacokinetic studies of vaccines and antiviral agents.	In general, 3D models lack vascular emulation. However, with microfluidic devices used in organ-on-a-chip, it is possible to simulate human vascular-endothelial dynamics.
Immunological response	It cannot resemble the cellular interactions with immune cells, such as infiltration of pro-inflammatory cells as occurs in tissues.	In general, the immune interaction in animal models cannot adequately reflect the human immune responses because of the differences between species.	The most advanced and complex models (i.e., organ-on-a-chip) based on co-cultures of multiple cell types can evaluate interactions with cells of the immune system and the cellular response to viral infections.
Ethics	A suitable alternative that can reduce animal testing but can also have ethical problems associated with primary cell and stem cell culture origins.	Many ethical concerns due to the animal suffering and international, national, and institutional regulation are applicable.	Like 2D cell cultures, the ethical considerations linked to stem cell origins must be followed.

**Table 2 biomedicines-09-00602-t002:** 3D models used to study SARS-CoV-2 infection and drug evaluation.

Model	Cell	Contributions
Spheroid	iPS	To evaluate tropism, the cytotoxic effect on cardiomyocytes and drugs such as remdesivir in SARS-CoV-2 infection [102].
Scaffold base	hAT2 cell	Identification of different states of infected cells through the progression of infection and phenotypic changes of cells induced by SARS-CoV-2 infection [103].
	HBMVEC cell	To assess the impact of SARS-CoV-2 on the blood–brain barrier associated with the neuropathology associated with COVID-19 [104].
Organotypic raft culture	Epithelial cells of the proximal airways	Heterogeneous respiratory tract infection, predominantly in hair cells. SARS-CoV-2 induces a cytopathic effect in infected cells and uninfected neighboring cells. Evaluated drugs like hydroxychloroquine [105].
Organoid	iPS	SARS-CoV-2 tropism. Pancreatic alpha and beta cells, liver organoids, cardiomyocytes, and neural cells are permissive to infection by the SARS-CoV-2 virus [106,107].
		To evaluate cell tropism in lung organoids and colonic organoids. Identification of SARS-CoV-2 entry inhibitors, such as mycophenolic acid and quinacrine dihydrochloride [108].
		To evaluate cellular tropism and neurotoxic effect of SARS-CoV-2 [109].
	Human ESC	Evaluate cellular tropism and infectivity of blood vessels and human tubular kidney cells.To assess the impact of human soluble angiotensin-converting enzyme 2 (hrsACE2) on SARS-CoV-2 infection [110].
	Primary intestinal epithelial stem cells	Cell tropism, enterocyte infectivity, and cellular changes through infection [111].
	hBEpC	To evaluate SARS-CoV-2 infection and the effect of drugs such as camostat on pulmonary organoids [112].
	Cholangiocytes	Cellular tropism and damage to liver tissue and bile ducts by SARS-CoV-2 [113].
Organ-on-a-chip	hAT2 HULEC-5a	SARS-CoV-2-induced lung injury may be mediated by communication between the epithelium–endothelium interface and immune cells. Remdesivir evaluation for infection [114].
	data HUVEC cell, Caco-2, HT-29 and PBMC	Evaluation of damage to the intestinal barrier caused by SARS-CoV-2 [115].

iPS—induced pluripotent stem cells; hAT2—human alveolar epithelial cells type II; HBMVEC—human brain microvascular endothelial cells; HUVEC—human umbilical vein endothelial cells; Caco-2—human colon adenocarcinoma cells; HT-29—grade II human colorectal adenocarcinoma cells; PBMC—human peripheral blood mononuclear cells; ESC—embryonic stem cells; hBEpC—normal human bronchial epithelial cells; HULEC-5a—pulmonary microvasculature cell line.

## Data Availability

Not applicable.

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
