# Peer review of "3D Cell Culture Models in COVID-19 Times: A Review of 3D Technologies to Understand and Accelerate Therapeutic Drug Discovery"

_biomedicines, 2021, doi:10.3390/biomedicines9060602_

Round 1

Reviewer 1 Report

In this review, Figueroa et al. summarized the development of 3D cell culture models in virus research and emphasized the contribution of 3D cell culture models in fighting COVID-19. The idea is clear. However, the organization of materials, discussion, and the author’s perspectives on the significance, current, and future development of 3D cell culture need further improvements. The reviewer recommends it for publication in this journal after revision.

1) I strongly suggest that the introduction part has to be improved by adding more comparison with other current technologies of virus-host interactions study, antiviral drugs, and vaccine development, and reorganizing chapter 1 and chapter 2. After the introduction of significance, I suggest adding an introduction of 2D culture, animal models, and other strategies with their advantages and disadvantages, and compare the 3D culture with them.

2) In chapter 3, the specific, unique merits (may include disadvantage) of each 3D culture strategy for host-virus interaction study should be clearly demonstrated, e.g line 114 why Spheroids also can be proper to research other 114 diseases? Line 121, what’s hybrids 3D cell systems advantages for host-virus interaction study? Line 129, I am confused if only ECM-based scaffold is good for host-virus interaction study or scaffold based on other materials are also a potential choice? Line 158, why organotypic raft culture that recapitulates epithelial differentiation has the ability or is helpful for the study of those viruses? I just feel like the organ-on-a-chip is the best-written part in chapter 3 with a clear demonstration of the merits, the research summary as the supporting materials, and the thoughts/ perspectives from authors but still need a little bit of reorganization/improvement, like adding figures to help illustration. In general, I suggest that for each subgroup, start with the introduction of this 3D culture strategy, then follow with general application, specific merits or what makes it proper choice for host-virus interaction study, then follow with research summary, and end with perspectives from authors.

3) In chapter 4, line 251, the significance of 3D culture to the host-covid 19 studies should be illustrated more. At the end of the research summary, the authors’ perspectives should be presented. In addition, I suggest that the part from the introduction of the liver model to the end of 4.2 should be reorganized. I feel that a summary or discussion of the whole chapter 4 should be added at the end.

4) In chapter 5, I just feel that the discussion is a little bit weak. For example, line 342 “many aspects to be evaluated and improved” should be illustrated more to clearly indicate the aspects that needed improvement. And how do those improvements help to contribute to the fighting with covid-19?

5) At the beginning of each chapter, there is a Typesetting mistake.

Reviewer 2 Report

The authors review various 3D culture models used in basic research and drug discovery. The emphasis is given to models used in SARS-CoV-2 studies. The review is very relevant and timely and compiles SARS-CoV-2 research done using 3D models.

However, there are some major issues that need to be  addressed before the review can be considered for publication.

  1. The language needs a major revision (preferentially by a native English speaker). Unfortunately, at times, it is almost impossible to understand what the authors are trying to say.
  2. In the abstract the authors say: “ 3D cell culture models are systems that permit the emulation of major tissue characteristics such as the in vivo immunological responses.” However, this very important aspect is not discussed in the text.
  3. 3D models are claimed to even replace tests in animal models. For the readers to understand the relevance and relationship of 3D models vs. animal models (which are extensively used in COVID-19 drug development despite their limitations) the authors could provide a paragraph on the translational relevance of these 3D models.
  4. Fig1 figure legend could be written in more detail.
  5. In line 232: Definition of SARS-CoV-2 is inaccurate. There are non-structural proteins as well. It is highly recommended that the SARS-CoV-2 structure and life cycle is presented as a figure. In the same figure there could be presented the route of infection and the tissues affected by the virus.
  6. The authors mostly describe what 3D models has been used in understanding SARS-Cov-2 pathophysiology, but very little is written on how 3D culture models has been used in therapeutic drug discovery, especially in the case of COVID-19. This should be discussed more in the text.

Round 2

Reviewer 2 Report

The authors have adequately addressed my concerns. Thank you.